# Effect of the Landscape on Insect Pests and Associated Natural Enemies in Greenhouses Crops: The Strawberry Study Case

**DOI:** 10.3390/insects14030302

**Published:** 2023-03-21

**Authors:** Marianne Doehler, Delphine Chauvin, Anne Le Ralec, Émeline Vanespen, Yannick Outreman

**Affiliations:** 1UMR 1349 IGEPP, Institut Agro, Université Rennes 1, INRAE, 35000 Rennes, France; 2AOPn Fraises de France, 47310 Estillac, France; 3UMR 1349 IGEPP, Institut Agro, Université Rennes 1, INRAE, 35650 Le Rheu, France

**Keywords:** conservation biological control, landscape context, protected cultivation, strawberry crops, natural enemies of pests, crop colonization, insect pests, diversity

## Abstract

**Simple Summary:**

The surrounding environment of greenhouses presents habitats, refuges, shelters, or food sources for insects, representing sources of risk for crop colonization by insect pests. A growing number of studies confirm the spontaneous influx of pests and their natural enemies in greenhouses. Identifying the properties of the surrounding landscape that affect this greenhouse crop colonization would help to improve pest prevention and biological control methods. Here, we present our study on the effect of landscape on insect colonization in greenhouse strawberry crops over two seasons in the South West of France. Our results showed that the surrounding landscape has contrasting and specific effects on crop colonization by different pest and biological control agent groups. Furthermore, the degree of openness of a greenhouse and the pest management practices marginally modulated insect crop colonization, whereas seasonality was a key factor of greenhouse crop colonization by insects. The various responses of insect groups to the surrounding landscape support the idea that management methods should involve the surrounding environment.

**Abstract:**

Compared to open-field crops, the influence of the surrounding landscape on insect diversity in greenhouse crops has been poorly studied. Due to growing evidence of insect influx in greenhouses, identifying the landscape properties influencing the protected crop colonization by insect pests and their natural enemies would promote the improvement of both pest prevention and conservation biological control methods. Here, we present a field study on the effect of the surrounding landscape on the colonization of greenhouse crops by insect pests and associated natural enemies. By monitoring 32 greenhouse strawberry crops in the South West of France, we surveyed crop colonization by four insect pests and four natural enemy groups over two cultivation periods. Our results showed that the landscape structure and composition could have contrasting effects on insect colonization of greenhouse crops so there could be species-specific effects and not general ones. While the degree of openness of greenhouses and the pest management practices modulated insect diversity marginally, we also showed that seasonality represented a key factor in insect crop colonization. The various responses of insect pests and natural enemy groups to the landscape support the idea that pest management methods must involve the surrounding environment.

## 1. Introduction

Landscape ecology aims at understanding how both the composition and structure of the landscape modulate the ability of organisms to find and exploit the resources they need and influence the dynamics of populations and communities [1]. By affecting animal movements [2] and interactions among species [3], landscape properties can partly explain the colonization of crops by the aerial dispersal of insects and the regulation of their populations by natural enemies. The application of landscape ecology concepts to agroecosystems can thus lead to identifying innovative ways to control crop pests and favor beneficial organisms.

Since the 1950s, intensive farming has led to a great simplification of agricultural landscapes [4], with a significant decline in semi-natural habitats. This landscape homogenization created favorable conditions for crop-specialized insect pests. Additionally, intensive agricultural practices, such as the frequent use of pesticides, strongly reduced local biodiversity and its associated ecosystem services including the regulation of pest populations [5,6,7,8]. Developing strategies to minimize the conflict between crop yield and biodiversity conservation is essential to restore fundamental ecosystem services and limit the negative impacts of agriculture. In this way, understanding the ecology of the natural enemies of insect pests at the level of the surrounding landscape may favor pest population regulation while reducing the use of chemicals.

Because it relies on such knowledge, conservation biological control of insect pests is seen as a promising strategy for reducing insecticide use and increasing the sustainability of agricultural production systems [9,10]. This control strategy consists of modifying field environment or cultural practices to protect and enhance specific natural enemies (i.e., the Biological Control Agents, hereafter noted BCAs) to reduce the damages that pests make [11]. However, expanding one’s view to include the surrounding landscape seems necessary as insect pests and associated BCAs migrate across the landscape for various needs, and this can lead to landscape management for enhancing natural biological control. 

The impacts of landscape on insect diversity and ecosystem services in open-field crops were analyzed in a large number of studies [12,13,14,15]. The landscape complexity would often have a positive effect on the diversity, abundance, and activities of natural enemies of insect pests [11]. It is also widely accepted that non-crop habitats (e.g., woodland, fallow land, grassland, etc.) provide essential functions for a wide range of pests and BCAs and can serve as a biodiversity source [12,14,16,17,18]. Overall, the impacts of cultivated areas on insect abundance and biological control are less evident than those of non-crop habitats [14,19]. If the landscape structure can also influence pest suppression, the strength and direction of those effects depend on the context [15]. In general, both the direction and significance of the landscape effects on pests are less clear than for natural enemies [12,13,14,15].

Compared to open-field crops, the number of studies testing the impacts of the surrounding landscape on pests and BCAs in greenhouse crops is still scarce. If most insect pests colonize the protected crops through the transportation of plant materials [20], indigenous species can enter greenhouses through the openings (e.g., ventilation windows). These migration events from the surrounding environment may lead to higher pest diversity than expected [21]. Once an insect pest colonizes a greenhouse spontaneously, the crop system may offer excellent conditions for its development, leading to additional problems in pest management. On the other hand, the natural enemies of insect pests can also enter greenhouses, and then contribute to pest control [20]. By offering habitats, refuges, shelters, or food sources for insect pests and their natural enemies, the environment surrounding the greenhouses can influence pest dynamics in protected crops. Given these ecological and agricultural consequences, one should determine the significance and direction of the impacts of the surrounding landscape of greenhouses on the diversity of insect pests and BCAs. Up to now, only three published studies analyzed the landscape effects on insect diversity in greenhouses crops (Appendix A summarizes their main results). If these studies underlined the importance of the surrounding landscape on insect dynamics, they focused on BCAs only. To define relevant practices for reducing pest crop colonization or promoting their biological control by local BCAs, a joint analysis of the landscape effects on both pests and BCAs in greenhouses is necessary.

Our main goal was to demonstrate that the surrounding landscape is an essential driver of insect diversity colonizing greenhouse crops. For this purpose, we have considered strawberry crops in France as a protected crop model. In France, strawberry cultivation represents 75,000 t for 4000 ha in 2021 [22] and is sensitive to various insect pests that cause damage to plants, transmit diseases, and affect fruit quality [23,24]. In addition to pesticides used to control insect pests, growers often use augmentative biological control [25,26] but with insufficient results, especially concerning aphid parasitoid releases [27]. Recent studies showed that aphid communities in greenhouse strawberry crops could be highly diverse (i.e., up to 13 species) [19] and that parasitoid species entering greenhouse crops contributed more effectively to aphid control than those released [28]. To understand how the surrounding landscape contributes to the insect flows in greenhouse strawberry crops, we have tested the effect of several landscape metrics on the occurrence of various insect pests and BCAs on strawberry plants in 32 greenhouses during two sampling sessions. We hypothesized that (1) the surrounding landscape can influence the colonization of the greenhouse crops by both pests and BCAs, (2) this effect can differ among insects, regarding their ecological or life cycle features, (3) the insects’ influx can also vary according to the cultivation season and the degree of openness of greenhouses, and (4) the chemicals and biological treatments can mitigate the influence of the surrounding landscape on the presence of insects in greenhouses. From our results, we discussed the variability of insect responses to the landscape and research perspectives, which may promote the development of new pest management methods such as conservation biological control.

## 2. Materials and Methods

### 2.1. Monitored Greenhouses

The most important strawberry production basin in France was considered in this study: the South-West of France. In this region, growers use various types of greenhouses to produce strawberries that differ notably by their opening: the closed greenhouses consist of plastic greenhouses, glass greenhouses, or high tunnels with insect-proof nets while open ones are high plastic tunnels without insect-proof devices. Thirty-two greenhouses were selected for the present study and to test the effect of the degree of openness of greenhouses on the studied ecological variables, the sampled greenhouses varied in their type (i.e., open or closed) (see Figure 1 and Appendix A for details). The monitored greenhouses were located in three different French departments of the South-West region: Lot-et-Garonne (27 greenhouses), Dordogne (4 greenhouses), and Gironde (1 greenhouse). These departments present contrasting cultivation conditions in terms of climate and landscape context. The minimum distance between the two sampled greenhouses was 750 m while the maximum distance was 122 km. In all studied greenhouses, strawberries are grown in rows in a soilless substrate and plants originated from nurseries. Early-season cultivars were planted in late autumn to early winter and started to produce fruits during the spring, while everbearing cultivars were planted in late winter and produced fruits from spring to early autumn. In each monitored greenhouse, we recorded the insect pest management practices used: the use of insecticides and/or the release of BCAs (e.g., releases of aphid predators, aphid parasitoids, or thrips predators). Appendix A summarizes the pest management practices used in the monitored greenhouses. Almost all growers used insecticide treatments and released thrips predators (i.e., *Orius* sp., *Neoseiulus cucumeris*, *Amblyseius swirskii*).

### 2.2. Insect Sampling

The dominant insect pests colonizing protected strawberry crops and some of their associated natural enemies were sampled in the monitored greenhouses. Data were collected during the spring (from April to May 2021) and the late spring-the early summer (in June 2021). As far as was possible, we sampled the same greenhouses across sampling sessions and in the same temporal sequence. As strawberry cultivation in four initial greenhouses ended in June, we considered four new greenhouse crops at the second sampling session.

The objective of the sampling was to identify the presence of insect pests and their associated enemies in the protected strawberry crops. For this purpose, we randomly selected between 20 and 50 strawberry plants distributed throughout each of the monitored greenhouses. On each strawberry plant, the presence/absence of the following dominant pest groups was noted: aphids, thrips, phytophagous bugs, and whiteflies. Although fruit flies are relevant insect pests, they were not monitored because their survey requires dedicated sampling techniques (i.e., fly traps). For the natural enemies of insect pests, we noted the presence/absence of the following BCA groups: aphid predators (i.e., ladybugs, hoverflies, and lacewings), aphid mummies (i.e., aphids parasitized by hymenopteran parasitoids), predatory thrips (i.e., *Aeolothrips* sp.) and predatory bugs (i.e., *Orius* sp., *Anthocoris* sp., and *Macrolophus* sp.). Given our sampling technique, predatory mites were not surveyed due to their small size. From these samplings, we obtained estimates of the ratio of plants colonized by each studied biological group in a given greenhouse (i.e., the number of plants colonized by a studied group divided by the total number of plants sampled). 

### 2.3. Landscape Description

Land-cover maps of the sampled greenhouses were obtained from aerial digital maps provided by Géoportail (https://www.geoportail.gouv.fr, accessed on 12 April 2021). For each monitored greenhouse, the landscape structure was estimated in a 500 m radius circular sector by using QGIS version 3.16 [29]. The land covers were noted during the insect samplings and classified into nine general categories, including cereal crop, legumes crop, oleaginous crop, vegetable and red fruit crop, orchard, semi-natural habitat/grassland, woodland, water and urban. Water represented the wetland, ponds and lakes, and urban represented human architecture and roads. The percentage of each category was measured within each landscape circle. To estimate the landscape connectivity, the length of hedges was calculated via the “length” function of QGIS. The number of patches (i.e., the contiguous area comprising a single land cover type, e.g., a forest or crop field [15]), their average surface, and the number of land cover types were also extracted for each mapping. Finally, to characterize the landscape complexity, a Shannon diversity index was calculated using the percentage of each land cover category. Table 1 summarizes the landscape metrics considered for their effects on insect colonization in greenhouse strawberry crops.

### 2.4. Data Analysis

All analyses were performed using the statistical program R version R 4.2.2 [30]. Firstly, the variation of the surrounding landscape was inspected by performing a Principal Component Analysis (PCA) including all the landscape metrics recorded by using FactoMineR package [31]. The level of landscape diversity and the most discriminating landscape metrics were identified from this descriptive analysis. Secondly, we analyzed the eight responses: the ratio of sampled plants colonized by aphids, phytophagous bugs, whiteflies, thrips, aphid predators, predatory thrips, or predatory bugs and the ratio of sampled plants presenting aphid mummies in the greenhouse strawberry crops. Each response was analyzed against the sampling session (i.e., a two-level fixed factor), the degree of openness of greenhouses (i.e., a two-level fixed factor), and all recorded variables describing the surrounding landscape (i.e., 14 continuous explanatory variables) by using generalized linear mixed models (GLMM), assuming a binomial error and using a logit-link function. Pest management practices were also included in the model. Because almost all growers used insecticide treatments and released predators of thrips (i.e., 30 and 31 greenhouse crops out of the 32 studied, respectively), these two practices were not included in the models. In the aphid occurrence model, we included the two following two-level fixed factors: the release of aphid predators (i.e., ladybirds, hoverflies, lacewings, *Aphidoletes aphidimyza*) and the release of aphid parasitoids. In the aphid mummies presence model, we included the release of aphid parasitoids as a fixed factor. In our dataset, the statistical unit was a strawberry plant. As some plants originated from the same greenhouse crop, the identity of the monitored greenhouse was included in the model as a random factor to account for data dependency.

Before the statistical modeling, we checked for multicollinearity (i.e., the correlation between explanatory variables) by calculating the VIF (Variable Inflation Factors) values among continuous explanatory variables using usdm package [32]. A cut-off VIF value of 3 was used to remove collinear variables [33]. For each GLMM, we used a backward selection model procedure (i.e., from the full model to the model containing only significant covariates) based on the significance of the model terms. The term significance was estimated using a likelihood ratio test. As we performed multiple tests (i.e., one model per each response), the *p*-values were corrected using Benjamini–Hochberg False Discovery Rate correction [34]. From the candidate model, the parameter estimates associated with each significant term were analyzed to interpret the fitted models. GLMM analyses were conducted using the lme4 package [35]. Finally, we calculated for each GLMM the marginal R^2^ (variance explained by the fixed effects) and the conditional R² (variance explained by both fixed and random effects) [36], using the usdm R package [32] The contribution of random effects (i.e., the greenhouse crop) can be deduced by subtracting the marginal R² from the conditional R².

## 3. Results

### 3.1. Variation in the Landscape Surrounding the Monitored Greenhouses

The two first components of the PCA describing the surrounding landscape accounted for 25.7% and 15.8% of total inertia, respectively (Figure 2). The wide spread of data points suggests that the thirty-two greenhouses monitored presented contrasting surrounding landscapes. This high variation was a prerequisite for our study on the effects of the surrounding landscape on crop colonization by insects. The first principal component has large associations with the number of patches, the number of cover types, and the mean patch surface, so this component primarily refers to landscape complexity. The second component has large associations with the percentage of oleaginous crops, percentage of urban areas, and percentage of semi-natural habitats and grasslands, so this component primarily measures the composition of the surrounding landscape. Hence, the variance between the recorded landscapes implied various landscape properties (i.e., complexity, composition, and connectivity). 

### 3.2. Landscape and Insect Occurrences

Overall, we inspected the presence/absence of insect pests and their associated natural enemies on 1785 strawberry plants. Among the studied insect pests, aphids were the most frequent in the monitored greenhouse crops: during the first session, aphids were detected in all monitored greenhouses while 93% of the sampled greenhouses were infested with aphids during the second sampling session. From the first and the second sampling session, thrips infested 62% and 83% of the monitored greenhouse crops, whitefly 64% and 86% and phytophagous bugs, 28% and 31%, respectively. Considering the aphids’ natural enemies, mummified aphids were detected in 62% and 93% of the monitored greenhouses. While lacewings were the most observed aphid predators in greenhouse crops during the first session (38% of the monitored greenhouses), 45% of those crops presented ladybugs during the second sampling session. For the other BCAs monitored, 5.2% and 1.6% of the 1785 plants inspected presented predatory bugs and predatory thrips, respectively. Because of these very low occurrences, these two BCA groups were not studied further. Table 2 details the average frequencies of strawberry plants colonized by a studied species group in the monitored greenhouse crops. 

From the analysis of multicollinearity considering the VIF values, we excluded from all GLMMs the two following covariates: the average patch surface and the percentage of cereal crops in the surrounding landscape (see Appendix A). For all insect pest groups and the natural enemies of aphids, the sampling session factor had a significant effect: all species groups presented the highest ratios of crop colonization during the second session (Table 2 and Table 3). The degree of openness of greenhouses influenced the ratio of crop colonization by some studied insect groups (Table 3, Figure 3). The ratio of plants colonized by aphids was significantly lowest in open greenhouses (open 34%; closed 66%) and highest in greenhouses where aphid predators have been released. Aphid parasitoid activity in greenhouses depended also on the degree of openness, in that the ratio of crop plants presenting aphid mummies was lowest in open greenhouses (open 31%; closed 61%). The release of aphid parasitoids in greenhouses did not influence the occurrence of aphids nor the presence of aphid mummies in crops. Interestingly, the landscape metrics influenced crop colonization by pests and beneficial insects in a variable manner (Figure 3). The ratio of strawberry plants colonized by aphids varied positively according to the percentage of semi-natural habitats and grasslands in the surrounding landscape. For the natural enemies of aphids, the ratio of crop plants presenting mummified aphids declined with the percentage of urban areas in the landscape while landscape metrics have no impact on the crop colonization by aphid predators. The ratio of plants colonized by whitefly increased with the number of patches and decreased with the percentage of orchards in the surrounding landscape. Finally, the ratios of strawberry plants colonized by thrips varied according to different landscape metrics: while these ratios declined with the number of land cover types, they increased with the percentage of woods (Table 3). Overall, calculations of both conditional R² and marginal R² for each GLMM suggest that our models explained between 28% and 65% of the responses’ variance. The fixed effects and the random effect (i.e., the identity of the monitored greenhouse) had quite similar contributions to the variance explanation (Table 3).

## 4. Discussion

In 2000, the total world area covered by greenhouses was nearly 300,000 ha [25]. By offering conditions that maximize crop yield per surface unit, the area under protected cultivation has recently increased in agriculture worldwide [37]. Despite the massive expansion of greenhouse crops, little is known about the main factors determining the occurrence of insect pests and their natural enemies in the protected crops. There is no doubt that insects can enter greenhouses by different routes, and the contribution of the local environment to spontaneous inflows needs to be clarified. While the literature is abundant on the effect of landscape on insect diversity in open-field crops (i.e., over a thousand studies published since 2000), only three articles analyzed the relationship between landscape metrics and insect diversity in greenhouse crops. Using the greenhouse strawberry crops as model, we highlight that the landscape surrounding the greenhouse crop could significantly contribute to crop colonization by insect pests and beneficial insects.

In many studies carried out on open-field crops and in the three published works conducted on greenhouse crops, authors focused on natural enemies. Here, we considered four dominant groups of strawberry pests and four groups of natural enemies. Results showed that the presence of aphids is widespread in almost all greenhouse crops in the South West of France, as observed by Postic et al. [21]. Thrips and whiteflies were also very frequent, phytophagous bugs being the less prevalent pests. We also confirm that aphid parasitoids and, to a lesser extent, aphid predators (i.e., ladybugs, hoverflies, lacewings) commonly colonized greenhouses spontaneously [21]. In contrast, predatory bugs and predatory thrips were rarely observed. Since most growers have released predatory bugs to control pest populations, the low prevalence of these BCAs is very surprising and implies a very low ability to maintain themselves in strawberry crops. Beyond these general observations, our first hypothesis that the surrounding landscape could influence the colonization of greenhouses crops by insect pests and BCAs is partly confirmed. Interestingly, the landscape properties influencing crop colonization varied among the monitored insect groups, especially between pests and BCAs. 

The colonization of greenhouse strawberry crops by aphids was positively influenced by the percentage of semi-natural habitats and grasslands in the surrounding landscape. In contrast, for open-field wheat crops, Alignier et al. [38] found that the abundance of aphids was negatively correlated with the proportion of grasslands, whatever the buffer size (200 m, 500 m and 1200 m). Alignier et al. [38] also showed that hedges in the landscape surrounding the wheat crops influence aphid parasitism positively but inversely, woods have a negative influence on parasitoids’ aphid control. Here, the presence of aphid mummies in the greenhouse strawberry crops declined according to the proportion of urban areas. Aphid parasitoids are poor dispersers [39], the effect of urban areas on the landscape could be explained by the low availability of floral resources required by parasitoids. The surrounding landscape had in this case no effect on aphid predator occurrences in greenhouse crops, in contrast to the open-field studies where positive effects of natural habitats or woods were observed [12,13,38]. Contrasts between our results and the open-field results are not strictly explained by agricultural systems; monitored aphid species and natural enemies varied and have different ecological needs. Due to the great diversity in aphid and BCA species colonizing strawberry crops [21], encompassing specialist and generalist species at each trophic level, a better estimate of the landscape effects on the crop colonization by these species would be achieved by considering each species separately and as far as possible each trophic chain, as proposed by [40]. This implies collecting a larger amount of data for each aphid species and their associated natural enemies in order to demonstrate more robust landscape effects.

In the monitored greenhouse crops, the colonization by thrips was negatively influenced by the landscape complexity (i.e., the number of land use types) but positively by its composition (i.e., the proportion of woodlands). Complex landscapes being favorable for the abundance of thrips predators [41], a fine-grained landscape would expose thrips to higher predation. The positive woodland influence on thrips crop colonization, although weak (Figure 3), could be due to the additional nutritional resources they provide such as hornbeam and ivy [42]. In contrast, in open-field crops, forests are supposed to act as barriers against thrips or as sources of natural enemies [43]. A survey of thrips natural enemies in greenhouses would be necessary to disentangle the various effects of landscape structure on pest thrips. As for aphids, it is necessary to identify phytophagous thrips at the species level, as some of them are specific to greenhouse crops while others can use several wild plants in the surrounding landscape.

The colonization of strawberry crops by whiteflies was positively influenced by landscape complexity (i.e., the number of patches) but negatively by the proportion of orchards. Strawberry whiteflies such as *Aleyrodes lonicerae* being polyphagous, the positive effect of landscape complexity could be related to additional resources provided by various landscape components. However, due to the lack of ecological knowledge on strawberry whiteflies, no valuable explanation for the orchards effect could be proposed. Finally, phytophagous bugs are not influenced by any landscape metrics although effects of the proportion of fallows and other herbaceous semi-natural cover in the landscape have been found on predatory bugs of the genus *Macrolophus* in greenhouses [44,45].

Our study also aimed at testing whether the cultivation season and the degree of openness of a greenhouse are determining factors in crop colonization by insects. First, we showed that all insect groups studied were significantly influenced by the sampling period: the colonization of the crop was higher during the second session scheduled in summer. Although this result is not surprising for poikilothermic organisms, it confirms that by offering constant high temperatures, greenhouses are very favorable environments for insect pests and BCAs. Secondly, the influence of the degree of openness of greenhouses on the presence of pest insects and BCAs was surprisingly not so frequent and contradictory to the parsimonious hypothesis of a positive link between colonization and crop-landscape connectivity [46]. Indeed, we found that both crop colonization by aphids and the presence of aphid mummies were negatively influenced by the degree of openness. Postic et al. [21] showed that the degree of openness of a greenhouse strawberry crop influences aphid density in different ways: while *A. malvae* and *R. porosum* are more frequent in closed greenhouses, *A. gossypii* is more frequent in open ones. This variation could be explained by the varying colonization routes among species (e.g., spontaneous influx through openings, transportation of plant materials, accidental introductions…). Concerning the activity of aphid parasitoids, the low number of mummified aphids in open greenhouses could be explained by the reduced number of aphids in such crop systems: parasitoid population dynamics are often coupled to those of its hosts, so a density-dependence effect could be assumed [47]. Here, we did not consider aphid hyperparasitoids. However, the hyperparasitism rates in French strawberry greenhouses can be locally high and higher in open greenhouses than in closed ones [21]. Those organisms should, therefore, be considered in future works as they can disrupt biological control by parasitoids.

One objective of our study was also to test whether the chemicals and biological treatments can mitigate the influence of the surrounding landscape on the presence of insects in greenhouses. Because almost all the growers use insecticides during strawberry cultivation, it was not possible to explain any variation in the presence of the different insect groups by this practice. Regarding the release of beneficial insects, the presence of aphids was positively influenced by the release of aphid predators. This contradictory result can be explained by predators releasing in the greenhouses where aphids were abundant. This result, together with the absence of the effect of parasitoid releases on aphid occurrence, tends to confirm the low efficiency of some commercial natural enemies [47].

By considering a multi-species approach rarely used before, we showed that the probability of the presence of a pest or a beneficial insect on a strawberry plant varied according to the properties of the surrounding landscape, the season, and, to a lesser extent, the degree of openness of the greenhouse. Taxa we studied would present contrasting characteristics ranging from very generalist to more specialist pest species; for natural enemies, a specialist with parasitoids and generalists with the set of generalist aphid predators (e.g., ladybirds, hoverflies, lacewings). The monitored insects also vary in terms of dispersal abilities (e.g., plant bugs within a few hundred meters, thrips within several kilometers) and life-cycle requirements (e.g., hoverflies have a variable diet depending on their stage of development whereas thrips have a wide range of hosts available and may be less constrained by their environment). In open-field crops, landscape effects can change according to ecological specialization, i.e., generalist insects respond on a larger scale than specialists. It would, therefore, be interesting to consider the ecological traits of the insects encountered on strawberry crops as they could also explain the variability of the observed effects in greenhouses. Indeed, the model explained only 20% to 50% of the variance of responses, with the random effect (i.e., the greenhouse studied) contributing half of this explained variance. So, other factors related to the local environment or crop-specific characteristics contribute to the colonization dynamics by insects in a greenhouse. Recent studies showed that the role of plant diversity near greenhouses may have effects on colonization by pests and their natural enemies (e.g., [20,48]). Additionally, agricultural practices may explain a large part of the insect community present in greenhouses (e.g., growing conditions of the young plants, prophylactic measures, pesticide use, the release of biocontrol agents, fertilization [24,49,50]), but we need a dataset with more variable practices for testing this hypothesis, especially regarding the insecticide uses. The next steps will aim to specify the landscape effects on the main pests and BCAs in strawberry crops, at different spatial scales including the adjacent environment ([20]), and at evaluating interactions between environmental effects and agricultural practices, which differ deeply in greenhouses compared to open-field crops. The consideration of such covariates should enhance our understanding of crop colonization by pests and their regulation by BCAs and the development of practices limiting pest entries and favoring BCAs by conservation biological control methods.

## Figures and Tables

**Figure 1 insects-14-00302-f001:**
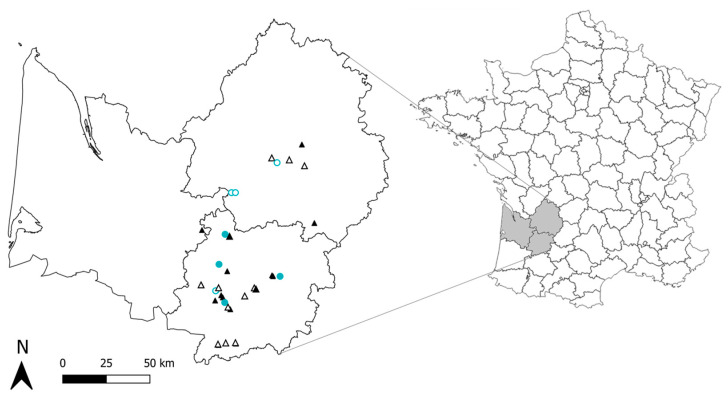
Location of the study area in France. Some sites being very close to each other, they are merged on the map. Black triangles represent greenhouse crops sampled at the two sampling sessions and blue circles represent greenhouse crops sampled only once. Empty symbols represent open greenhouses and solid symbols represent close greenhouses.

**Figure 2 insects-14-00302-f002:**
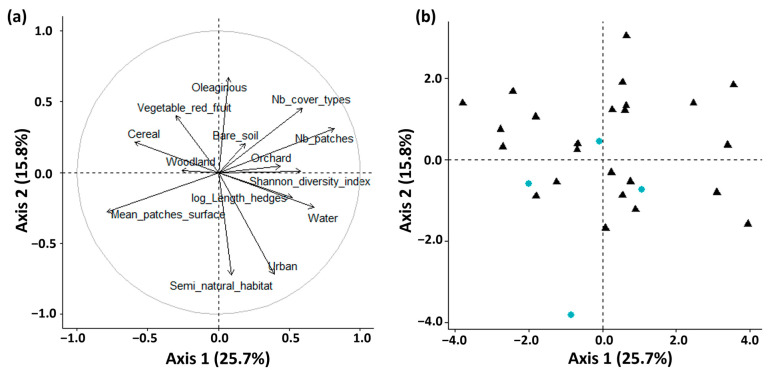
Principal component analysis (PCA) on the landscape metrics around the 32 greenhouse strawberry crops monitored in the present study (see Table 1 for details). (**a**): Variable vectors map in dimensions 1 and 2; (**b**) Individuals (i.e., recorded greenhouses) position in dimensions 1 and 2. Black triangles represent greenhouse crops sampled at the two sampling sessions (first session and second session), blue circles represent greenhouse crops sampled only once (second session).

**Figure 3 insects-14-00302-f003:**
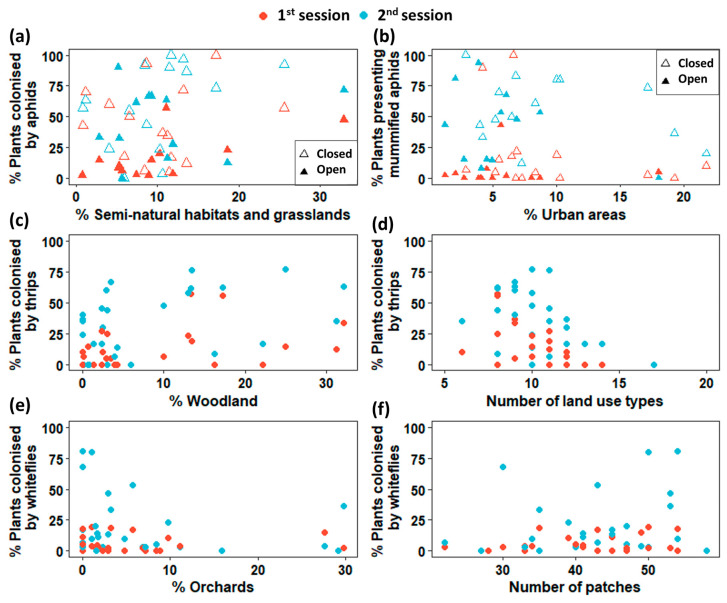
Effect of various landscape metrics on the colonization of greenhouse strawberry crops by pest and beneficial insects. (**a**) Percentage of plants colonized by aphids according to the percentage of semi-natural habitats and grasslands; (**b**) Percentage of plants presenting mummified aphids according to the percentage of urban areas; (**c**) Percentage of plants colonized by thrips according to the percentage of woodland; (**d**) Percentage of plants colonized by thrips according to the number of land use types; (**e**) Percentage of plants colonized by whiteflies according to the percentage of orchards; (**f**) Percentage of plants colonized by whiteflies according to the number of land use types. Red point represents the surveys in the first sampling session (April–May) and blue point in the second sampling session (June). Empty triangles represent closed greenhouses and solid triangles represent open greenhouses. When degree of openness of the greenhouses had no significant effect (see text and Table 3), points are represented by circles.

**Table 1 insects-14-00302-t001:** List of landscape metrics considered for their effects on insect colonization within greenhouse strawberry crops.

Landscape Properties	Landscape Metric	Metric Components	Maintained after VIF ^1^ Analysis
Landscape composition	% Cereal crops	Corn, Wheat, Oats, Barley	**NO**
	% Oleaginous crops	Sunflower, Rapeseed	yes
	% Vegetable and red fruit crops	Potato, Beetroot, Sweet Pepper/Chili, Tomato, Eggplant, Raspberry, Blueberry, Red fruit, Zucchini, Salad, Strawberry	yes
	% Orchards	Walnut, Hazelnut, Chestnut, Orchard, Kiwi, Vine, Forestry, Tree nurseries	yes
	% Semi-natural habitats and grassland	Grassland, Weedy area, Wasteland, Flower strips	yes
	% Woodland	Wood	yes
	% Water	Rivers, Reservoirs, Ponds, Basin	yes
	% Bare ground		yes
	% Urban	Residential areas, Urban environment, Industrial zones	yes
Landscape heterogeneity	Shannon diversity index	H’= −Σ *p_i_* ln *p_i_*; *p_i_* = the percentage of the land cover category *i*	yes
	Number of land cover types		yes
Landscape connectivity	log (Length of hedges)		yes
Landscape fragmentation	Number of patches		yes
	Mean patch surface		**NO**

^1^ VIF-values: Variance Inflation Factors.

**Table 2 insects-14-00302-t002:** Ratio of strawberry plants in greenhouse crops colonized by a given insect group according to the sampling session (mean ± standard error).

Sampling Session	Aphids	Phytophagous Bugs	Thrips	Whiteflies	Aphid Parasitoids	Aphid Predators
1st Session	0.317 ± 0.054	0.014 ± 0.005	0.129 ± 0.031	0.052 ± 0.012	0.021 ± 0.009	0.127 ± 0.048
2nd Session	0.516 ± 0.062	0.049 ± 0.021	0.337 ± 0.048	0.198 ± 0.046	0.085 ± 0.019	0.494 ± 0.058

**Table 3 insects-14-00302-t003:** Effects of the landscape metrics, sampling session, the degree of openness of greenhouses and pest management practices on the insect crop colonization in greenhouse strawberry crops (Generalized Linear Mixed Models). Significance of the explanatory variables was corrected by the Hochberg–Benjamini FRD. β: coefficient estimates of the significant covariate; se: standard error of estimates. Conditional R^2^ refers to the contribution of both fixed and random effects to the response variance explained by the model. Marginal R² refers to the contribution of fixed effects to the response variance explained by the model.

Response	Significant Explanatory Variables	β	se (β)	*p*-Value of β	Conditional R^2^	Marginal R^2^
% Plants infested with aphids	% Semi-natural habitat and grassland	0.061	0.020	0.003	0.379	0.258
	Session 1st	0	0.000			
	Session 2nd	1.638	0.126	<0.001		
	Closed greenhouse	0	0.000			
	Open greenhouse	−1.158	0.318	<0.001		
	No release of aphid predators	0	0.000			
	Release of aphid predators	0.950	0.381	0.012		
% Plants infested with phytophagous bugs	Session 1st	0	0.000		0.637	0.067
	Session 2nd	1.555	0.388	<0.001		
% Plants infested with thrips	% Wooded area	0.055	0.024	0.030	0.503	0.293
	Number of land use types	−0.421	0.121	0.001		
	Session 1st	0	0.000			
	Session 2nd	1.503	0.150	<0.001		
% Plants infested with whiteflies	Number of patches	0.081	0.001	0.020	0.499	0.223
	% Orchards	−0.078	0.001	0.028		
	Session 1st	0	0.000			
	Session 2nd	1.712	0.001	<0.001		
% Plants infested with aphid predators	Session 1st	0	0.000		0.282	0.136
	Session 2nd	1.582	0.001	<0.001		
% Plants presenting mummified aphids	% Urban area	−0.130	0.061	0.037	0.648	0.377
	Session 1st	0	0.000			
	Session 2nd	3.044	0.188	<0.001		
	Closed greenhouse	0	0.000			
	Open greenhouse	−2.325	0.631	<0.001		

## Data Availability

Data are available as Appendix A (Data sheets Landscape data).

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
