# Peer review of "Effect of the Landscape on Insect Pests and Associated Natural Enemies in Greenhouses Crops: The Strawberry Study Case"

_insects, 2023, doi:10.3390/insects14030302_

Round 1
Reviewer 1 Report
This study presents the results on 1 year sampling in 32 strawberry greenhouses in France and shows how pest and natural enemy abundance is related to the environment. In addition, this study includes an overview of the recent literature about landscape effects on pest management in open field and greenhouse crops. Combining these two things is in my opinion a bit too much, although I understand “Insects” has no limitations in MS lengths. The literature overview is useful, but is does not really address new insights based on these studies. Hence, I think it is for the reader better to reduce the literature review in size and included it in the introduction for the field study.
The field study is well analysed and presented, but I have a major concern about the influences of agricultural practices.
It is not analysed or included at all how the activities of the growers related to pesticide use or releases of natural enemies could have effected the results. I think that most growers either would have applied pesticides or released natural enemies, such as predatory mites for the control of thrips and parasitoids for the control of whiteflies and aphids. These factors should be included in the analyses as a factor or it should be explained and argued that these practices varied limited among the sampled greenhouses and are ignored.
In the insect sampling I also miss some crucial analyses. First, hyperparasitoids are rather common and their presence was not checked in the insect sampling. This could have been monitored by checking the parasitoids emergence holes of aphid mummies were (different between primary and secondary parasitoids). So the presence and abundance of hyperparasitoids could also have affected the abundance of primary parasitoids and aphids, which is now ignored. Second, I do not understand why thrips and whiteflies were not identified to species level. This is rather important because some species are typical for greenhouses and do not occur at all or only in low numbers outside, like Frankliniella occidentalis. Thus for these species you would not expect a large influx from the greenhouse surrounding. If this is not known, it might be better to exclude the data from the analyses.
Author Response
Reviewer 1
Comments and Suggestions for Authors
This study presents the results on 1 year sampling in 32 strawberry greenhouses in France and shows how pest and natural enemy abundance is related to the environment. In addition, this study includes an overview of the recent literature about landscape effects on pest management in open field and greenhouse crops. Combining these two things is in my opinion a bit too much, although I understand “Insects” has no limitations in MS lengths. The literature overview is useful, but is does not really address new insights based on these studies. Hence, I think it is for the reader better to reduce the literature review in size and included it in the introduction for the field study.
Author response:
The literature review was reduced and mainly included in the introduction part.
The field study is well analysed and presented, but I have a major concern about the influences of agricultural practices.
It is not analysed or included at all how the activities of the growers related to pesticide use or releases of natural enemies could have effected the results. I think that most growers either would have applied pesticides or released natural enemies, such as predatory mites for the control of thrips and parasitoids for the control of whiteflies and aphids. These factors should be included in the analyses as a factor or it should be explained and argued that these practices varied limited among the sampled greenhouses and are ignored.
Author response:
We agree that the use of chemical pesticides as well as the releases of beneficial insects could have significant effect on the presence of insects in greenhouses. In fact, the data concerning the chemical and biological treatments were collected during the insect sampling. The proportion of each type of pest management method used in the 32 monitored greenhouses is given in the Supplementary Table 2. The pest management practices were then included in the statistical analyses. Unfortunately, almost all growers used insecticides during cultivation so that this effect could not be tested. Consideration of the pest management practices in the study led to a major rewrite of the article (cf. Materials and methods, L 143 to 148 and 209 to 215; Results, L. 278 to 291; Discussion, L 418 to 426).
In the insect sampling I also miss some crucial analyses. First, hyperparasitoids are rather common and their presence was not checked in the insect sampling. This could have been monitored by checking the parasitoids emergence holes of aphid mummies were (different between primary and secondary parasitoids). So the presence and abundance of hyperparasitoids could also have affected the abundance of primary parasitoids and aphids, which is now ignored.
Author response:
We agree with this comment. A previous study we conducted (Postic et al. 2020) showed that the hyperparasitism rates in French strawberry greenhouses can be locally high and these rates can be higher in open greenhouses than in closed ones. Those organisms should therefore be considered in future works, as they would have negative effects on biological control by parasitoids. We included this point in the revised version (L 413-417)
Second, I do not understand why thrips and whiteflies were not identified to species level. This is rather important because some species are typical for greenhouses and do not occur at all or only in low numbers outside, like Frankliniella occidentalis. Thus for these species you would not expect a large influx from the greenhouse surrounding. If this is not known, it might be better to exclude the data from the analyses.
Author response:
During our sampling, we identified at the species level both aphids and phytophagous bugs because we have the requested skills. However, information at the species level is not useful given our goals and the results concerning species mentioned in the previous version were only informative. Therefore, in order to treat all insect groups in the same way and to avoid confusion in our objectives, we have removed all references to species identification. For thrips and whiteflies, we mention now that it is necessary to identify phytophagous thrips at the species level, as some of them are specific to greenhouse crops while others can use several wild plants in the surrounding landscape (L 382-384).
Reviewer 2 Report
This paper combines a limited literature review on landscape effects on pest and natural enemy populations in greenhouses with the analysis of a sampling campaign in strawberry greenhouses in the south of France.
I have made several comments in a pdf of the article, which I have attached. There are some problems with the English writing that have to be addressed. As I am not a statistician, I can't judge the statistical analyses.
Here are my main reservations:
- The authors appear to be entirely oblivious of the fact that certain pest management interventions in these commercially exploited greenhouse strawberry crops may have affected population densities of the main pests and their natural enemies (but see their comment in parentheses in line 519). I have not seen any information in the paper whether pest management interventions were carried out in the crops sampled by the authors. Were chemical pesticides being used in any of these greenhouses and couldn't they have differentially affected certain pest and beneficial insects there? Were any natural enemy releases being done throughout the crop cycle? Were any of the parasitoids and predators perhaps the result of previous augmentative releases, rather than immigrants from surrounding landscape? Such interventions will obviously affect pest and natural enemy levels in the crop. This information should ideally be collected and mentioned for each of the greenhouses sampled, as it may have had stronger effects on the insect levels in the crop than any of the landscape features studied!
- Sampled insect groups were not treated equally. Why were bugs and aphids identified to species level and thrips and whiteflies were not? Can the authors provide more information on the species of lacewings, ladybirds and hoverflies encountered?
- Figure 2 is missing from the submission

Author Response
Reviewer 2
Comments and Suggestions for Authors
This paper combines a limited literature review on landscape effects on pest and natural enemy populations in greenhouses with the analysis of a sampling campaign in strawberry greenhouses in the south of France.
Author comment:
We would like to thank the Reviewer #1 for his well-founded comments. By considering all these comments, the paper has been improved significantly.
I have made several comments in a pdf of the article, which I have attached. There are some problems with the English writing that have to be addressed. As I am not a statistician, I can't judge the statistical analyses.
Author response:
All of reviewer 1’s comments done in the pdf file were considered in the revised manuscript. In addition, the article was proofread by Owen Cleary, an English native speaker. We hope that there are no more problems with English writing
Here are my main reservations:
- The authors appear to be entirely oblivious of the fact that certain pest management interventions in these commercially exploited greenhouse strawberry crops may have affected population densities of the main pests and their natural enemies (but see their comment in parentheses in line 519). I have not seen any information in the paper whether pest management interventions were carried out in the crops sampled by the authors. Were chemical pesticides being used in any of these greenhouses and couldn't they have differentially affected certain pest and beneficial insects there? Were any natural enemy releases being done throughout the crop cycle? Were any of the parasitoids and predators perhaps the result of previous augmentative releases, rather than immigrants from surrounding landscape? Such interventions will obviously affect pest and natural enemy levels in the crop. This information should ideally be collected and mentioned for each of the greenhouses sampled, as it may have had stronger effects on the insect levels in the crop than any of the landscape features studied!
Author response:
We agree that the use of chemical pesticides as well as the releases of beneficial insects could have significant effect on the presence of insects in greenhouses. In fact, the data concerning the chemical and biological treatments were collected during the insect sampling. The proportion of each type of pest management method used in the 32 monitored greenhouses is given in the Supplementary Table 2. The pest management practices were then included in the statistical analyses. Unfortunately, almost all growers used insecticides during cultivation so that this effect could not be tested. Consideration of the pest management practices in the study led to a major rewrite of the article (cf. Materials and methods, L 143 to 148 and 209 to 215; Results, L. 278 to 291; Discussion, L 418 to 426).
- Sampled insect groups were not treated equally. Why were bugs and aphids identified to species level and thrips and whiteflies were not? Can the authors provide more information on the species of lacewings, ladybirds and hoverflies encountered?
Author response:
Our main goal was to test whether the surrounding landscape can influence the colonization of greenhouse strawberry crops by the dominant insect pest and natural enemy groups. During our sampling, we identified at the species level both aphids and phytophagous bugs because we have the requested skills. However, information at the species level is not useful given our goals and the results concerning species mentioned in the previous version were only informative. Therefore, in order to treat all insect groups in the same way and to avoid confusion in our objectives, we have removed all references to species identification.
- Figure 2 is missing from the submission
Author response:
We would like to apologize for this error. This figure has been included in the revised manuscript.
Reviewer 3 Report
More recently, greenhouses have been widely concerned by urban agriculture. In contrast to open field crops, very little is known about the role of the surrounding landscape on insect diversity in greenhouse crops. The manuscript studies on the effect of landscape on insect pests and natural enemies in greenhouse strawberry crops in the South West of France. The work is interesting and meaningful. The experiment design is reasonable and the language is fluent. More importantly, the results of the manuscript support the idea that pest management methods must imply the surrounding environment. In general, the manuscript is well organized. I’d like to recommend this paper to publish in your journal after a minor review.
Below are minor comments:
1. Whether the title can be considered?’state of art’ is not the point. In my opinion, attention is the strawberry study case.
2. The literature review about the landscape effects on insect pests and natural enemies diversity in open-field crops and greenhouse crops is very comprehensive. However, it may be better to split it in the Introduction or Discussion.
3. In 2.1. Materials and Methods, I didn't see whether chemical control treatment is needed. Chemical control is a very important factor for insect pests and natural enemies.
4. Part of the information in Figure 1 and Table 1 is repetitive. Is it possible to put Table 1 in the supplementary materials, and redraw a clear and beautiful map including a variety of experimental settings of monitored greenhouses.
5. Table 1. Description of the 32 monitored greenhouse strawberry crops. What does the v and NA mean? Please give a comment after the table.
6. L270-271: There are many kinds of insect pests on strawberry including aphids, thrips, whiteflies, fruit fly, spider mites and other small insects . Please inform why you choose these insects and how to identify the specific species of aphids, thrips and whiteflies.
7. L272-273: Please inform why you only choose focused on biological control agents of aphids. I am a little confused that how to determine the parasitoids and predators only target aphids.
8. L327\L328\L346: 4 should be an obvious sequence error.
9. Please supply Figure 2.
10. L350-352: I suggest to supply the proportion of different aphid species, especially three most observed aphid species.
11. Strawberry has many diseases and insect pests. Is there any relationship between diseases and pests? Whether it can be explained in the discussion?
12. The reference formats does not fit the requirements of the journal, please check carefully.
Author Response
Reviewer 3
Comments and Suggestions for Authors
More recently, greenhouses have been widely concerned by urban agriculture. In contrast to open field crops, very little is known about the role of the surrounding landscape on insect diversity in greenhouse crops. The manuscript studies on the effect of landscape on insect pests and natural enemies in greenhouse strawberry crops in the South West of France. The work is interesting and meaningful. The experiment design is reasonable and the language is fluent. More importantly, the results of the manuscript support the idea that pest management methods must imply the surrounding environment. In general, the manuscript is well organized. I’d like to recommend this paper to publish in your journal after a minor review.
Author comment:
We would like to thank the Reviewer #2 for his positive and well-founded comments. By considering all these comments, the paper has been improved significantly.
Below are minor comments:
- Whether the title can be considered? ’state of art’ is not the point. In my opinion, attention is the strawberry study case.
Author response:
We changed the title accordingly.
- The literature review about the landscape effects on insect pests and natural enemies diversity in open-field crops and greenhouse crops is very comprehensive. However, it may be better to split it in the Introduction or Discussion.
Author response:
The literature review was reduced and mainly included in the introduction part.
3. In 2.1. Materials and Methods, I didn't see whether chemical control treatment is needed. Chemical control is a very important factor for insect pests and natural enemies.
Author response:
We agree that the use of chemical pesticides as well as the releases of beneficial insects could have significant effect on the presence of insects in greenhouses. In fact, the data concerning the chemical and biological treatments were collected during the insect sampling. The proportion of each type of pest management method used in the 32 monitored greenhouses is given in the Supplementary Table 2. The pest management practices were then included in the statistical analyses. Unfortunately, almost all growers used insecticides during cultivation so that this effect could not be tested. Consideration of the pest management practices in the study led to a major rewrite of the article (cf. Materials and methods, L 143 to 148 and 209 to 215; Results, L. 278 to 291; Discussion, L 418 to 426).
- Part of the information in Figure 1 and Table 1 is repetitive. Is it possible to put Table 1 in the supplementary materials, and redraw a clear and beautiful map including a variety of experimental settings of monitored greenhouses.
Author response:
This figure is considered now as supplementary material (i.e. Supplementary Figure 1)
- Table 1. Description of the 32 monitored greenhouse strawberry crops. What does the v and NA mean? Please give a comment after the table.
Author response:
Table has been modified accordingly
- L270-271: There are many kinds of insect pests on strawberry including aphids, thrips, whiteflies, fruit fly, spider mites and other small insects . Please inform why you choose these insects and how to identify the specific species of aphids, thrips and whiteflies.
Author response:
Our main goal was to test whether the surrounding landscape influences the colonization of strawberry crops by the dominant groups of insect pests. In South West of France, aphids, thrips, bugs and whiteflies are the more problematic pests. In fruit flies appeared during the summer, these pests were not monitored because their survey requires dedicated sampling techniques (i.e. flies traps).
During our sampling, we identified at the species level both aphids and phytophagous bugs because we have the requested skills. However, information at the species level is not useful given our goals and the results concerning species mentioned in the previous version were only informative. Therefore, in order to treat all insect groups in the same way and to avoid confusion in our objectives, we have removed all references to species identification.
- L272-273: Please inform why you only choose focused on biological control agents of aphids. I am a little confused that how to determine the parasitoids and predators only target aphids.
Author response:
For the natural enemies of insect pests, we noted the presence/absence of the four following groups: aphid predators (i.e. ladybugs, hoverflies, and lacewings), aphid mummies (i.e. aphids parasitized by hymenopteran parasitoids), thrips predators (i.e. Aeolothrips sp.) and predatory bugs (i.e. Orius sp., Anthocoris sp., and Macrolophus sp.). As predatory bugs and thrips predators were rarely observed, these two groups were not studied further. Consideration of these two groups of natural enemies is now specified in the revised manuscript (cf. Materials and methods, L 168 to 172; Results, L. 264 to 268; Discussion, L 346 to 347).
- L327\L328\L346: 4 should be an obvious sequence error.
Author response:
Text has been changed accordingly
- Please supply Figure 2.
Author response:
We would like to apologize for this error. This figure has been included in the revised manuscript.
- L350-352: I suggest to supply the proportion of different aphid species, especially three most observed aphid species.
Author response:
As said before, in order to treat all insect groups in the same way and to avoid confusion in our objectives, we have removed all references to species identification.
- Strawberry has many diseases and insect pests. Is there any relationship between diseases and pests? Whether it can be explained in the discussion?
Author response:
Indeed, plant disease may interact with insects but this has not been considered here. We would like to thank the Reviewer #2 for this comment as we are still working on strawberry crop colonization by insects and plant disease will be now monitored.
- The reference formats does not fit the requirements of the journal, please check carefully.
Author response:
Reference format has been changed accordingly
Round 2
Reviewer 2 Report
I believe that after consideration of the reviewers' comments the paper has now been improved. However, I do have a few minor additional comments:
- the authors use "thrips predators" for both "predators of thrips" (e.g. line 148) and "predatory thrips" (e.g. line 171): this is confusing. I recommend they use "predatory thrips" whenever they refer to thrips that are natural enemies of other arthropods.
- it is surprising that predatory bugs were rarely found (line 265) whereas some of these (Orius) were in fact released in the crop by the growers (line 148): how do the authors explain this?
- Table 3: harmonize the annotation of the symbol "+/-" (now it is written in different styles)
Reviewer 3 Report
This is my second review of this manuscript, and I can say it is improved significantly after considering of the comments of the reviewers. I think the manuscript is overall well written and the topic is relevant in the field of “Insects Ecology and Biocontrol Applications”. So, I recommend the manuscript to be suited for publication.
The following are a few minor additional comments: in this manuscript:
1. L72-75: The impacts of landscape on insect diversity and ecosystem services in open-field crops were analyzed in a large number of studies (i.e. 1080 studies published from 2000 to 2022 using "landscape AND pest AND control AND insect" wording expression in Web of ScienceTM). Different reviews and meta-analyses have synthesized these studies [12– 15].
I think it is understandable to delete this paragraph.
2. L151: Generally, figure is more displaying and intuitive than the table. So can you redraw a detail-rich map including a variety of experimental settings of monitored greenhouses?
3. L203: “thrips predators” and “ predatory thrips”. These are not the same at all. It is confusing. It is clear to explain please.
4. L452: (see [20] for a review) can be directly replaced with “[20]”.
